ⓐ | **Open Peer Review** | Host-Microbial Interactions | Research Article

# Pharmacological inhibition of host pathways enhances macrophage killing of intracellular bacterial pathogens

**Ramesh Rijal,**[1,2] **Richard H. Gomer**[2]

**ABSTRACT** After ingestion into macrophage phagosomes, some bacterial pathogens such as *Mycobacterium tuberculosis* (*Mtb*) evade killing by preventing phagosome acidification and fusion of the phagosome with a lysosome. *Mtb* accumulates extracellular polyphosphate (polyP), and polyP inhibits macrophage phagosome acidification and bacterial killing. In *Dictyostelium discoideum*, polyP also inhibits bacterial killing, and we identified some proteins in *D. discoideum* that polyP requires to suppress the killing of ingested bacteria. Here, we find that pharmacological inhibition of human orthologues of the *D. discoideum* proteins, including P2Y1 receptors, mammalian target of rapamycin, and inositol hexakisphosphate kinase, enhances the killing of *Mtb*, *Legionella pneumophila*, and *Listeria monocytogenes* by human macrophages. *Mtb* inhibits phagosome acidification, expression of the proinflammatory marker CD54, and autophagy and increases expression of the anti-inflammatory marker CD206. In *Mtb*-infected macrophages, the polyP-degrading enzyme polyphosphatase (ScPPX) and inhibitors reversed these effects, with ScPPX increasing CD54 expression more in female macrophages compared to male macrophages. In addition, *Mtb* inhibits proteasome activity, and some, but not all, inhibitors reversed these effects. While the existence of a dedicated polyP signaling pathway remains uncertain, our findings suggest that pharmacological inhibition of select host proteins can restore macrophage function and enhances the killing of intracellular pathogens.

**IMPORTANCE** Human macrophages engulf bacteria into phagosomes, which then fuse with lysosomes to kill the bacteria. However, after engulfment, pathogenic bacteria such as *Mycobacterium tuberculosis*, *Legionella pneumophila*, and *Listeria monocytogenes* can block phagosome-lysosome fusion, allowing their survival. Here, we show that pharmacological inhibition of specific macrophage proteins reverses these effects and enhances bacterial killing. These findings suggest that targeting host factors involved in these processes may provide a therapeutic strategy to improve macrophage function against infections such as tuberculosis, Legionnaires' disease, and listeriosis.

**KEYWORDS** polyphosphate signaling, macrophage signaling pathways, intracellular pathogens, host-pathogen interactions, host-directed therapies

Macrophages engulf invading microorganisms into a membrane-bound compartment called the phagosome (1–3). The phagosome then acidifies and fuses with lysosomes to form a phagolysosome (4). Within the phagolysosome, hydrolytic enzymes, antimicrobial compounds, and reactive oxygen and nitrogen species kill the ingested pathogen (4, 5). Pathogens such as *Mycobacterium tuberculosis* (*Mtb*) can prevent the fusion of phagosomes with lysosomes in macrophages, thereby evading the bactericidal actions of the lysosome and enabling them to persist within macrophages (6, 7). *Listeria monocytogenes* (referred to as *Listeria* hereafter), the bacterium that causes listeriosis, escapes from the phagosome into the cytosol, thus avoiding killing in the

Address correspondence to Ramesh Rijal, Ramesh.Rijal@usm.edu.

R.R. and R.H.G. are inventors on a patent application for the use of MRS2279 and TNP for the treatment of tuberculosis and other bacterial diseases (Patent application no. PCT/US2024/013914).

See the funding table on p. 14.

phagolysosome (8). *Legionella pneumophila* (referred to as *Legionella* hereafter), the bacterium that causes Legionnaires' disease, employs secreted factors to manipulate host cell processes and evade immune detection, allowing it to establish replicative vacuoles within host cells (9). Determining how these bacteria prevent macrophages from killing them could reveal strategies to enhance macrophage bactericidal function.

Polyphosphate (polyP) is a linear chain of phosphate units found across all forms of life, from bacteria to humans, playing crucial roles in energy storage, stress response, and metabolism (10). PolyP kinase (PPK), a widely conserved bacterial enzyme, catalyzes the synthesis of polyP from ATP (11). PolyP levels are also affected by polyphosphatase (PPX), an enzyme that degrades polyP (12). Pathogenic bacteria lacking PPK, or having reduced PPK levels, exhibit defects in stress response, quorum sensing, growth, survival, and virulence (13–20). *Mtb* possesses PPK1 and PPK2 enzymes, which are absent in humans (21). Intracellular polyP is necessary for the survival of *Mtb* in host cells (18, 22–24), and deletion of PPK1 in *Mycobacterium smegmatis (Msmeg)* attenuates the survival of ingested *Msmeg* in human macrophages (25). In addition to intracellular polyP, *Mtb* and *Neisseria gonorrhoeae*, the bacteria that causes gonorrhea, have polyP in their envelopes, and this envelope-associated polyP appears to protect these bacteria from antimicrobials (19, 20, 26). We previously found that *Mtb* also accumulates extracellular polyP, and calculations suggested that the concentration of polyP in the space inside a phagosome and outside the *Mtb* cell would quickly rise (25). Treatment of *Mtb*-infected macrophages with recombinant PPX (ScPPX) reduced the *Mtb* burden in macrophages (25), suggesting that both intracellular and extracellular polyP potentiate the survival of *Mtb* in host cells. Bacterial polyP also impairs macrophage function by interfering with proinflammatory polarization and suppressing type I interferon responses, thereby promoting bacterial survival (27). Treatment of *Mtb* with gallein, a broad-spectrum PPK inhibitor (28–30), potentiates the ability of isoniazid, an anti-*Mtb* antibiotic, to inhibit *Mtb* growth in both *in vitro* culture and within human macrophages (26). Together, these findings suggest that bacterial polyP may modulate macrophage responses in ways that enhance bacterial survival.

PolyP inhibits phagosome acidification and lysosome activity in the eukaryotic microbe *Dictyostelium discoideum* and human macrophages (25). Exogenous polyP promotes the survival of bacteria in *D. discoideum*, and by examining this effect in known mutants, we found that polyP requires the mTOR complex protein Lst8, the inositol hexakisphosphate kinase I6kA, and the Rho-GTPase RacE to facilitate survival of ingested bacteria (25, 31). Based in part on the polyP pathway identified in *D. discoideum*, we tested pharmacological inhibitors targeting the human orthologs of these proteins and found that several of these compounds potentiate the ability of macrophages to kill a test *Mtb* strain, an attenuated ΔleuDΔpanCD auxotroph of H37Rv (hereafter referred to as *Mtb*), which is approved for use under Biosafety Level 2 (BSL-2) conditions (32), as well as other intracellular pathogens including *Listeria* and *Legionella*.

## RESULTS

### MRS2279 and TNP potentiate the ability of macrophages to kill Mtb, Legionella, and Listeria

In *D. discoideum*, a signal transduction pathway appears to mediate the ability of polyP to enhance the survival of ingested bacteria (31). To determine whether similar mechanisms operate in macrophages, we tested pharmacological inhibitors targeting both human homologs of *Dictyostelium* proteins required for polyP-mediated bacterial survival and other macrophage proteins reported to interact with polyP. Table 1 lists proteins that may mediate polyP sensing. In humans, the purinergic receptor P2Y1 senses polyP (33), and the drug MRS2279 inhibits this receptor (34, 35). The human Receptor for Advanced Glycation End products (RAGE) also senses polyP (33), and FPSZM1 inhibits RAGE (36). For Lst8, IP6K, RacE, and RhoA (proteins identified in the *D. discoideum* screen), there are human orthologs for which inhibitors have been found. Rapamycin inhibits

human mTORC1 (37), TNP inhibits human IP6K (38), MBQ167 inhibits the small GTPases Rac/Cdc42 (39), Rhosin and Y16 inhibit the small GTPase RhoA (40, 41). AdcB (UniProt: Q86KB1; another protein identified in the *D. discoideum* screen) does not have a clear human ortholog, but Alphafold/Foldseek analysis suggested that human thioredoxin-interacting protein is a structural homolog of *Dictyostelium* AdcB (42–44). Extendin-4 inhibits human thioredoxin-interacting protein (45, 46).

Granulocyte macrophage-colony stimulating factor (GM-CSF) causes monocytes to become pro-inflammatory macrophages, whereas macrophage CSF (M-CSF) causes monocytes to become anti-inflammatory macrophages (47). Human blood monocytes were cultured for 6 days with GM-CSF or M-CSF to generate macrophages, and these were allowed to phagocytose *Mtb*, in the presence of ScPPX or the drugs listed in Table 1, washed, treated with 200 µg/mL gentamicin (Sigma, St. Louis, MO, USA) to kill uningested bacteria, and then lysed at 4 hours or at 48 hours with a detergent that does not kill bacteria. The ScPPX or drugs were present in all the washes and throughout the experiment. The lysates were plated to measure the number of viable bacteria. We previously observed that ScPPX reduces the ingested *Mtb* burden in GM-CSF macrophages at 48 hours (25). In GM-CSF macrophages at 4 hours (relatively shortly after ingesting bacteria), 1,000 nM MRS2279 had no significant effect on the number of ingested *Mtb* that were still viable, but 1,000 nM TNP increased the number of *Mtb* (Fig. 1A). At 48 hours, 1,000 nM MRS2279 or 1,000 nM TNP reduced the number of viable ingested *Mtb* (Fig. 1B). At 4 hours, 10 µg/mL ScPPX, 1,000 nM FPSZM1, or 1,000 nM Rapamycin did not significantly alter the number of viable ingested *Mtb* (Fig. S1A), whereas at 48 hours all three reduced the number of viable ingested *Mtb* (Fig. S1B). At 48 hours, 1,000 nM MBQ167, 1,000 nM Rhosin, 1,000 nM Y16, or 1,000 nM Extendin-4 did not significantly affect the viability of ingested *Mtb* (Fig. S1C).

To determine if longer treatment times also decrease the viability of *Mtb*, we lysed cells at 72 hours. Compared to control, 100 or 1,000 nM MRS2279, and 10,100, or 1,000 nM of TNP reduced the number of viable ingested *Mtb* in GM-CSF macrophages (Fig. 1C). All concentrations of MRS2279, or TNP, also reduced the number of viable ingested *Mtb* at 72 hours in M-CSF macrophages (Fig. 1D).

To determine if the drugs have similar effects on ingested *Legionella* and *Listeria*, GM-CSF macrophages were allowed to phagocytose *Legionella* or *Listeria* in the absence or presence of MRS2279 or TNP, and the viability of ingested bacteria was tested at 48 hours. 100 or 1000 nM MRS2279 and 10, 100, or 1000 nM TNP reduced the number of viable ingested *Legionella* (Fig. 1E). Similarly, 10 or 1000 nM MRS2279, and 10, One-hundred or 1,000 nM TNP reduced the number of viable ingested *Listeria* (Fig. 1F). For all the above results, there were no significant differences between macrophages from male donors (blue symbols in graphs) and female donors (red symbols). Together, these data suggest that pharmacological inhibition of multiple macrophage proteins, identified based on a putative polyP-responsive pathway, potentiates the killing of ingested bacteria, supporting the idea that targeting host pathways can restore macrophage killing of intracellular pathogens.

**TABLE 1** Inhibitors for human proteins or human orthologs of *D. discoideum* proteins identified in Rahman et al. (31)

| *D. discoideum* protein | Target protein in human or bacteria | Inhibitor |
|---|---|---|
| | Purinergic receptor P2Y1 | MRS2279 |
| | RAGE | FPSZM1 |
| Lst8 | mTORC1 | Rapamycin |
| Inositol hexakisphosphate kinase (IP6K) | IP6K and inositol 1,4,5-trisphosphate 3-kinase | TNP |
| Rac1 (RacE) | Small GTPases Rac/Cdc42 | MBQ167 |
| RhoA (RacE) | Small GTPase RhoA | Rhosin |
| | Rho guanine nucleotide exchange factors | Y16 |
| AdcB | TXNIP – Thioredoxin-interacting protein | Extendin-4 |

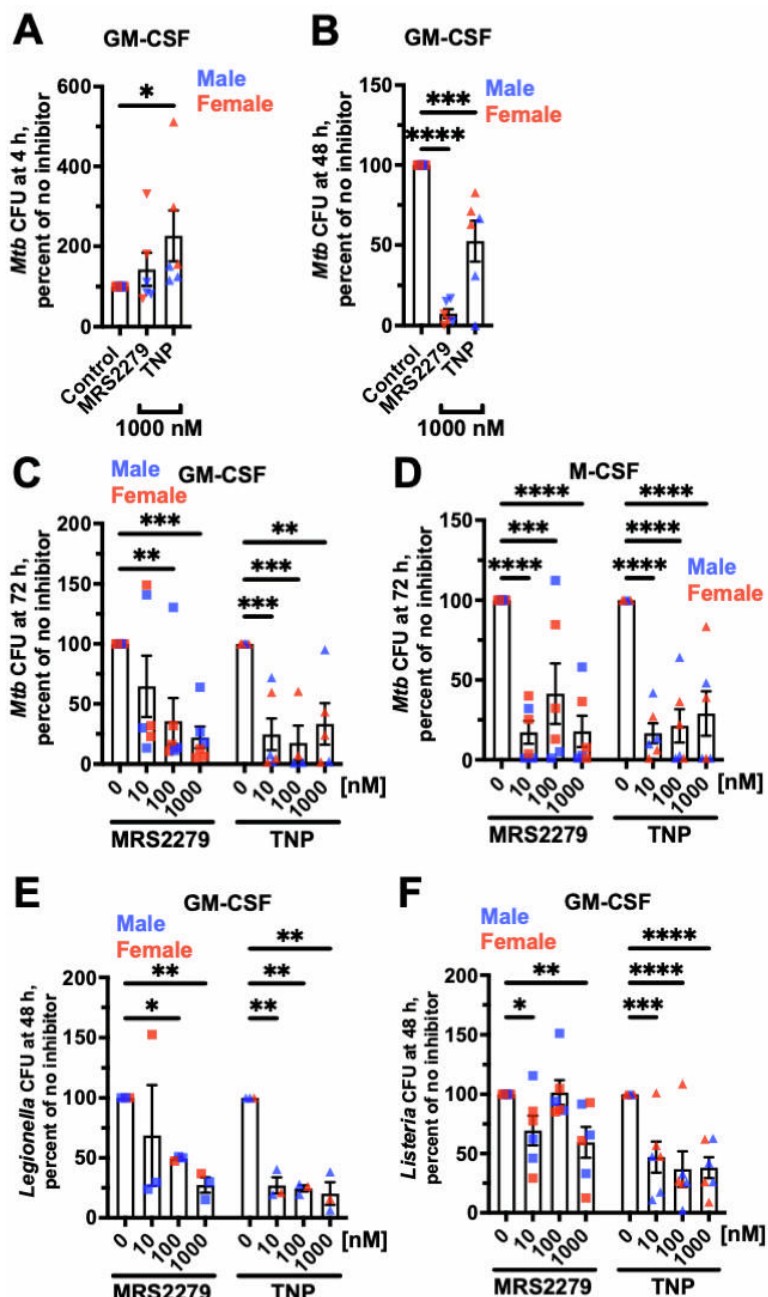

**FIG 1** MRS2279 (P2Y1 inhibitor) or TNP (IP6K inhibitor) potentiates the ability of macrophages to kill ingested *Mtb*, *Legionella*, or *Listeria*. (A–F) Viable ingested bacteria in macrophages with GM-CSF or M-CSF, in the absence (Control) or presence of the indicated concentration of MRS2279 and TNP, were determined as CFUs at 4 hours (A), 48 hours (B, E, and F), or 72 hours (C and D) after ingestion. For each experiment with each donor, CFU in the control was considered 100%. All values are mean ± SEM of six (three female and three male) (A–D and F) or three (one female and two male) (E) independent experiments. Male data points are shown in blue and female data points are shown in red. For each bar, there was no significant difference between male and female (unpaired *t*-test). *$P < 0.05$; **$P < 0.01$; ***$P < 0.001$; ****$P < 0.0001$ compared to the 0 control (one-way analysis of variance with Dunnett's test for A and B, and two-way analysis of variance with Dunnett's test for C–F).

## MRS2279 and TNP do not affect the viability of macrophages or uningested *Mtb*, Legionella, or Listeria

The viability of human macrophages can be tested by incubating cells with Deep Blue Cell Viability resazurin dye (48). Metabolically active cells reduce blue resazurin to the pink product resorufin (48). With this assay, none of the inhibitors significantly altered the metabolic activity of uninfected or infected macrophages (Fig. S2A through C). We also monitored the optical density, as a measure of growth, of *Mtb*, *Legionella*, or *Listeria* in the presence of 1,000 nM of the inhibitors with no macrophages present. None of the inhibitors tested significantly affected bacterial growth (Fig. S2D through F). Together, these results suggest that the inhibitors reduce the viability of ingested bacterial pathogens by enhancing macrophages' ability to kill the ingested bacteria and not by inhibiting macrophage viability or bacterial growth.

### MRS2279 and TNP block effects of exogenous polyP

Exogenous polyP inhibits the ability of *Dictyostelium* and human macrophages to kill ingested *Escherichia coli*, which do not accumulate detectable levels of extracellular polyP (25). To determine whether this effect of exogenous polyP can be reversed, GM-CSF macrophages were allowed to phagocytose *E. coli* in the absence or presence of 15 µg/mL exogenous polyP, with or without selected inhibitors. At 4 hours, polyP in the absence (no drug) or presence of MRS2279, FPSZM1, rapamycin, or TNP did not significantly affect the number of viable ingested *E. coli* (Fig. 2A). At 48 hours, polyP with no drug increased the number of viable ingested *E. coli*, and MRS2279, FPSZM1, rapamycin, and TNP partially prevented this effect (Fig. 2B). Possibly because ScPPX and polyP increased the number of ingested *E. coli* at 4 hours (Fig. 2A), ScPPX in the presence of exogenous polyP did not significantly decrease the viability of ingested *E. coli* at 48 hours (Fig. 2B). As above, there was no significant effect of blood donor sex on the results. These data suggest that the enhanced survival of ingested *E. coli* caused by exogenous polyP can be partially reversed by pharmacological inhibitors targeting host proteins that may mediate macrophage responses to polyP.

### MRS2279 and FPSZM1 prevent polyP-mediated inhibition of phagosome acidification

Bacterial pathogens, including *Mtb*, have evolved ways to prevent phagosome acidification (4, 7, 49), and exogenous polyP also inhibits phagosome acidification in human macrophages (25). To determine whether inhibitors could reverse this effect, GM-CSF or M-CSF macrophages were incubated with pHrodo red-labeled dead yeast particles. The pHrodo red-labeled particles (referred to as yeast hereafter) are non-fluorescent outside the cell but fluoresces brightly red in acidic phagosomes (50) (Fig. 2C; Fig. S3A). Concanamycin A (concanA), a vacuolar-type H+-ATPase inhibitor that blocks phagosome acidification (51), served as a positive control for inhibiting phagosome acidification (Fig. 2D through K). As previously observed (25), polyP reduced the fluorescence of ingested yeast in macrophages (Fig. 2D, E, H, I), and concanA also reduced the fluorescence of ingested yeast in macrophages (Fig. 2D, E, H, I). PolyP and concanA did not significantly affect the percentage of macrophages with yeasts or the number of ingested yeasts per macrophage (Fig. 2F, G, J and K). In the presence of polyP, 1,000 nM MRS2279 increased the fluorescence intensities of ingested yeast in GM-CSF and M-CSF macrophages (Fig. 2D, E, H, I), and 1,000 nM FPSZM1 increased the fluorescence intensities of ingested yeast in M-CSF macrophages (Fig. S3F and G). Neither polyP nor the inhibitors significantly altered the percentage of macrophages with ingested yeast (Fig. 2F and J; Fig. S3D and H) or the number of ingested yeasts per macrophage (Fig. 2G and K; Fig. S3E and I). Rapamycin or FPSZM1 did not significantly reverse the effect of polyP on the number of acidic phagosomes per cell in GM-CSF macrophages (Fig. S3B and C). There was no significant effect of blood donor sex on the results. Together, these data suggest that inhibitors such as MRS2279 and FPSZM1 can prevent polyP's effect on phagosome acidification.

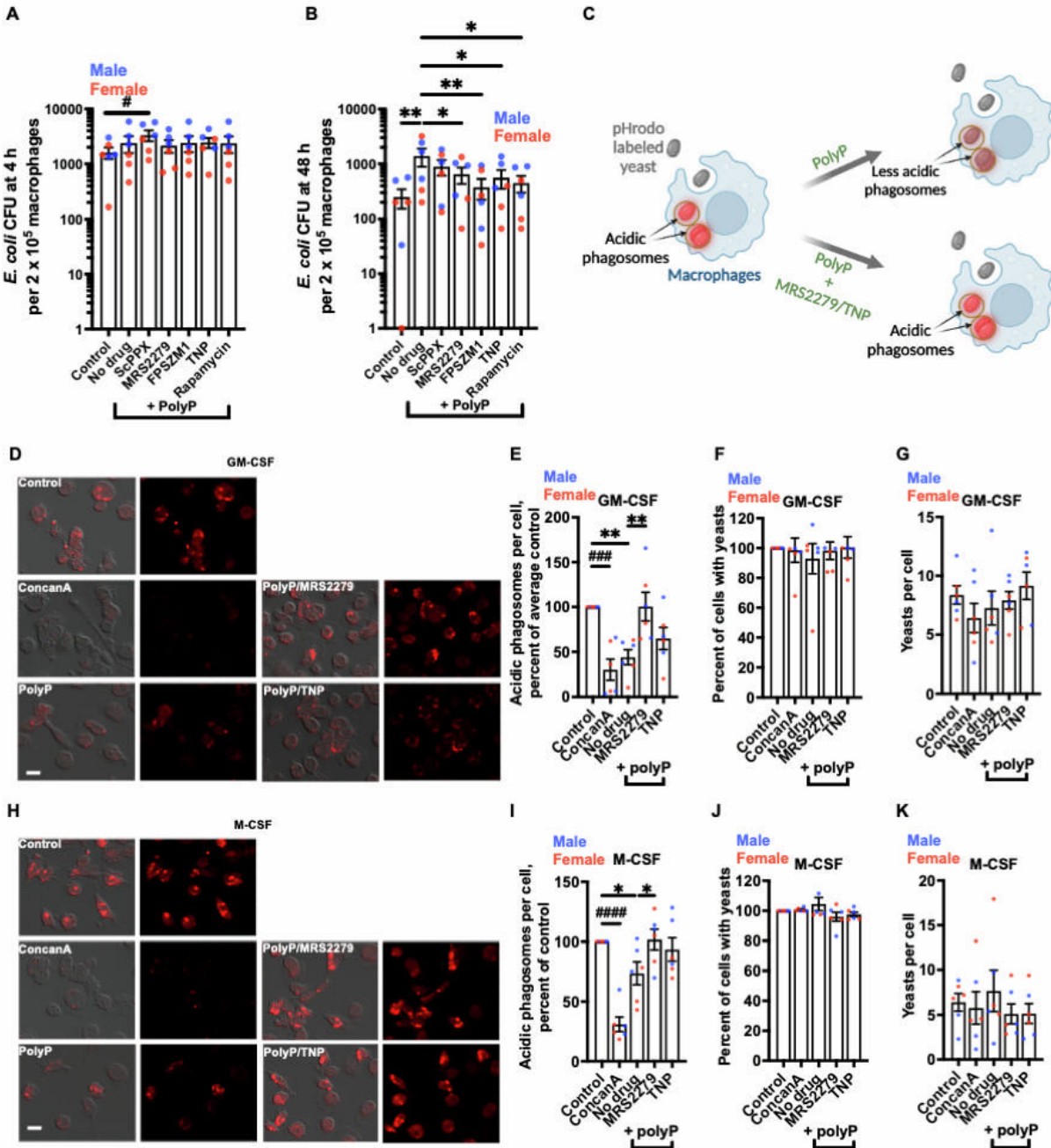

**FIG 2** Inhibitors prevent polyP-induced survival of ingested *E. coli* and reduced phagosome acidification. (A–B) Viable *E. coli* in GM-CSF macrophages, in the absence (Control) or presence of 15 µg/mL polyP, without (No drug) or with 1,000 nM of the indicated inhibitor, was determined as CFU at 4 hours (A),or 48 hours (B).CFU in all treatment conditions were compared with the No drug condition for statistical significance. (C) A schematic showing macrophages with phagosomes containing pHrodo red-labeled yeast, in the presence or absence of polyP, with or without MRS2279 (P2Y1 inhibitor) or TNP (IP6K inhibitor). pHrodo red-labeled yeast have low fluorescence outside the cell but show red fluorescence in acidic phagosomes. (D) Human GM-CSF macrophages were incubated with yeast, in the absence (Control) or presence of 15 µg/mL polyP without (No drug) or with 1,000 nM of the indicated inhibitor (+ polyP) for 1 hour, fixed, and fluorescence images were taken. 100 nM of ConcanA is a positive control for inhibition of phagosome acidification. DIC merged with fluorescence images is at the left, and fluorescence images are at the right for each treatment condition. Bar is 20 µm. Images are representative of six independent experiments (three females and three males). (E–G) Images from D were used to measure fluorescence intensities of yeast (E), percent of macrophages with yeasts (F), and number of yeasts per macrophages (G). The fluorescence intensity of yeast in control was set to 100 for E. (H–K) Experiments from D–G were performed with M-CSF macrophages. Size bars are 20 µm. Male data points are shown in blue and female data points are shown in red. For each bar, there was no significant difference between male and female (unpaired *t*-test). All values are mean ± SEM of six (three females and three males) independent experiments. * or # $P < 0.05$; **$P < 0.01$; ### $P < 0.001$; #### $P < 0.0001$ (one-way analysis of variance with Dunnett's test). * indicates compared to no drug; # indicates compared to control.

## Inhibition of polyP-related host responses enhances proinflammatory polarization

Enhanced expression of CD54, also known as intercellular adhesion molecule 1, is associated with the polarization of macrophages toward a proinflammatory phenotype with enhanced phagocytosis (52, 53). CD54 expression is decreased in *Mtb*-infected host cells (54). CD206, also known as the mannose receptor, is a cell surface pattern recognition receptor primarily expressed in anti-inflammatory macrophages and is involved in detecting and phagocytosing pathogens such as *Mtb* (55). Compared to the control, both polyP or *Mtb* alone reduced anti-CD54 staining and increased anti-CD206 staining of GM-CSF macrophages (Fig. 3A through D). The presence of 10 µg/mL ScPPX, or 1,000 nM MRS2279, FPSZM1, Rapamycin, or TNP in macrophage-*Mtb* coculture increased CD54 staining and decreased CD206 staining on macrophages (Fig. 3A through D), with ScPPX showing more of an effect on increasing CD54 expression in female macrophages compared to male macrophages. Together, these data suggest that except for ScPPX, all the inhibitors reduce the ability of *Mtb* to inhibit macrophage polarization away from a proinflammatory phenotype and reduce the ability of *Mtb* to increase CD206 expression in a sex-independent manner.

## Inhibitors restore proteasome activity suppressed by polyP or mtb

The proteasome generates peptides for antigen presentation by MHC Class I molecules (56, 57). *Mtb* inhibits antigen presentation and the induction of adaptive immune responses (58). We previously found that polyP inhibits proteasome activity in *Dictyostelium*, and this requires the mTOR protein Lst8, RacE, I6kA, the arrestin protein AdcB, and the polyP receptor GrlD (31). To determine if inhibitors affect proteasome activity, we measured proteasome activity in macrophages treated with polyP or infected with *Mtb* in the presence or absence of the inhibitors described above. Female macrophages showed increased basal proteasome activity compared to male macrophages (Fig. 4A). The proteasome inhibitor Mg132 (56) inhibited proteasome activity, and both polyP and *Mtb* reduced proteasome activity (Fig. 4B). The presence of 1,000 nM MRS2279 partially restored proteasome activity in *Mtb*-infected macrophages, whereas 10 µg/ml ScPPX, 1,000 nM FPSZM1, 1,000 nM TNP, or 1,000 nM rapamycin did not significantly alter proteasome activity (Fig. 4B). Together, these data suggest that *Mtb* may suppress macrophage proteasome activity via a mechanism involving polyP and that some host-targeted inhibitors can counteract this effect.

## DISCUSSION

In this report, we have shown that pharmacological inhibitors targeting the human polyP receptors P2Y1 or RAGE, and mTOR and I6KA, human orthologs of *Dictyostelium* proteins required for polyP-mediated bacterial survival, attenuate the survival of ingested *Mtb* in human macrophages, and inhibitors of P2Y1, and I6KA attenuate the survival of ingested *Legionella* or *Listeria*. These results suggest that some host responses to bacterial polyP may be conserved between *D. discoideum* and human macrophages, and that targeting these host components could offer a therapeutic strategy to enhance macrophage-mediated clearance of intracellular pathogens. However, the observed effects may not be exclusively mediated through P2Y1, RAGE, or I6KA, and the precise mechanism by which these inhibitors enhance macrophage antimicrobial activity remains to be fully elucidated.

*Listeria* secretes listeriolysin, which forms pores in the phagosome, and this allows the bacteria to move from the phagosome to the cytosol shortly after infection (59). Possibly because of this, the viability of *Listeria* appeared to be relatively less affected compared to *Mtb* and *Legionella* in the presence of MRS2279 or TNP.

Although polyP significantly reduces phagosome acidification in macrophages, our data show that this effect can be partially reversed by MRS2279 and, in M-CSF macrophages, by FPSZM1. However, the extent of restoration was modest, and the response

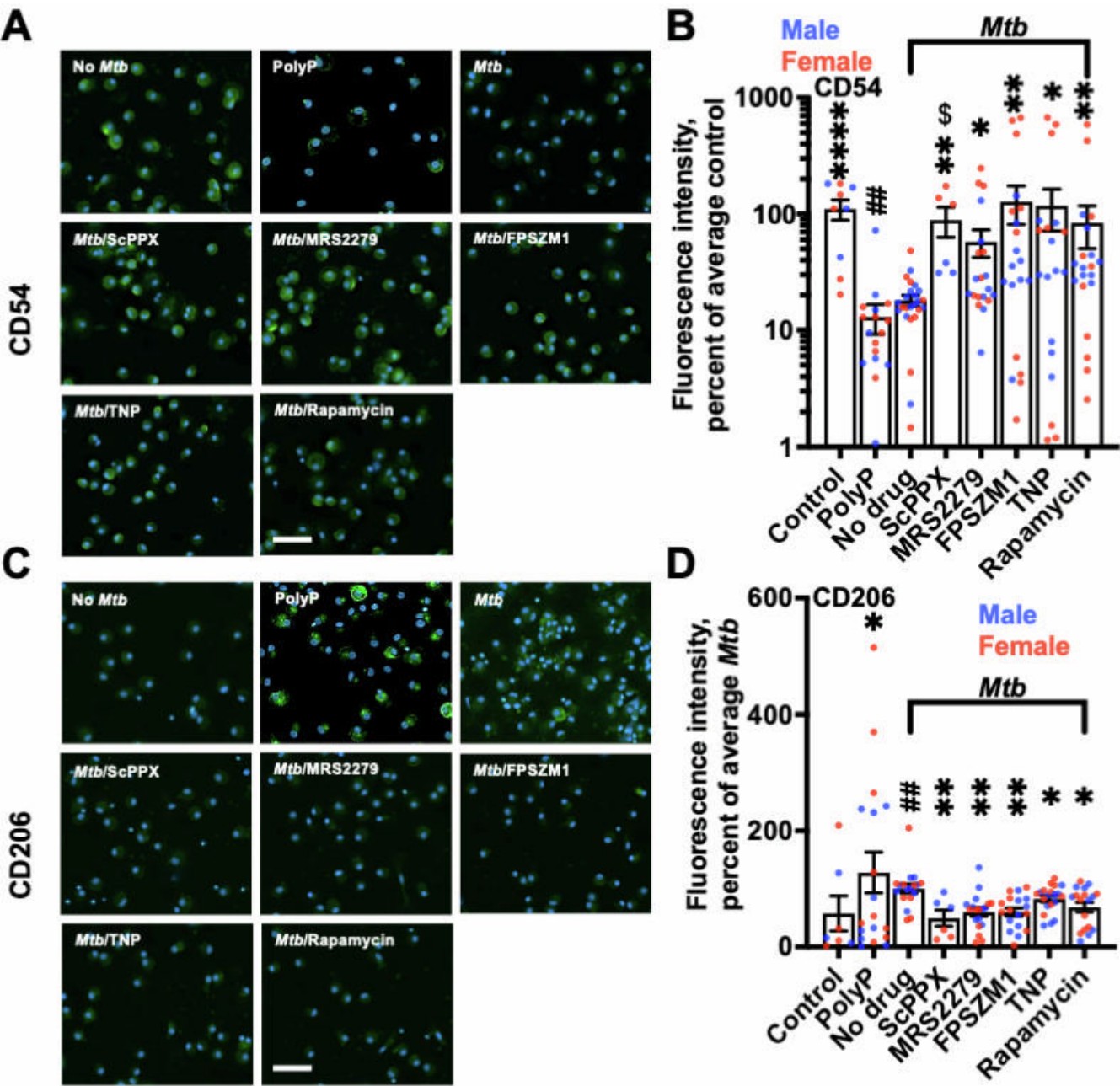

**FIG 3** Some inhibitors enhance the proinflammatory polarization of macrophages. Uninfected (control), 15 µg/mL polyP-treated, or *Mtb*-infected macrophages with GM-CSF in the absence (No drug) or presence of 10 µg/mL ScPPX or 1,000 nM of the indicated inhibitor at 24 hours after phagocytosis (as described in the bacterial survival assay) were fixed, permeabilized, and stained with antibodies against CD54 (A) or CD206 (C), and the mean fluorescence intensity per cell was measured and average of control (B) or No drug (D) was considered 100%. Bars are 100 µm. Fluorescence images are representative of six independent experiments (three female and three male). All values are mean ± SEM of six (three females and three males) independent experiments. One or more than one images from each independent experiment were analyzed, indicated by multiple data points. Male data points are shown in blue and female data points are shown in red. An unpaired *t*-test was performed to assess male vs. female differences for each bar in the graph. *$P < 0.05$; ** or ## $P < 0.01$; ****$P < 0.0001$ (Kruskal-Wallis test). * indicates compared to no drug; # indicates compared to control. $ $P < 0.05$ indicates female vs male (unpaired *t*-test).

varied slightly with macrophage polarization state. TNP did not rescue phagosome acidification despite partially reversing polyP-induced intracellular bacterial survival. One explanation may be that TNP influences downstream immune functions rather than

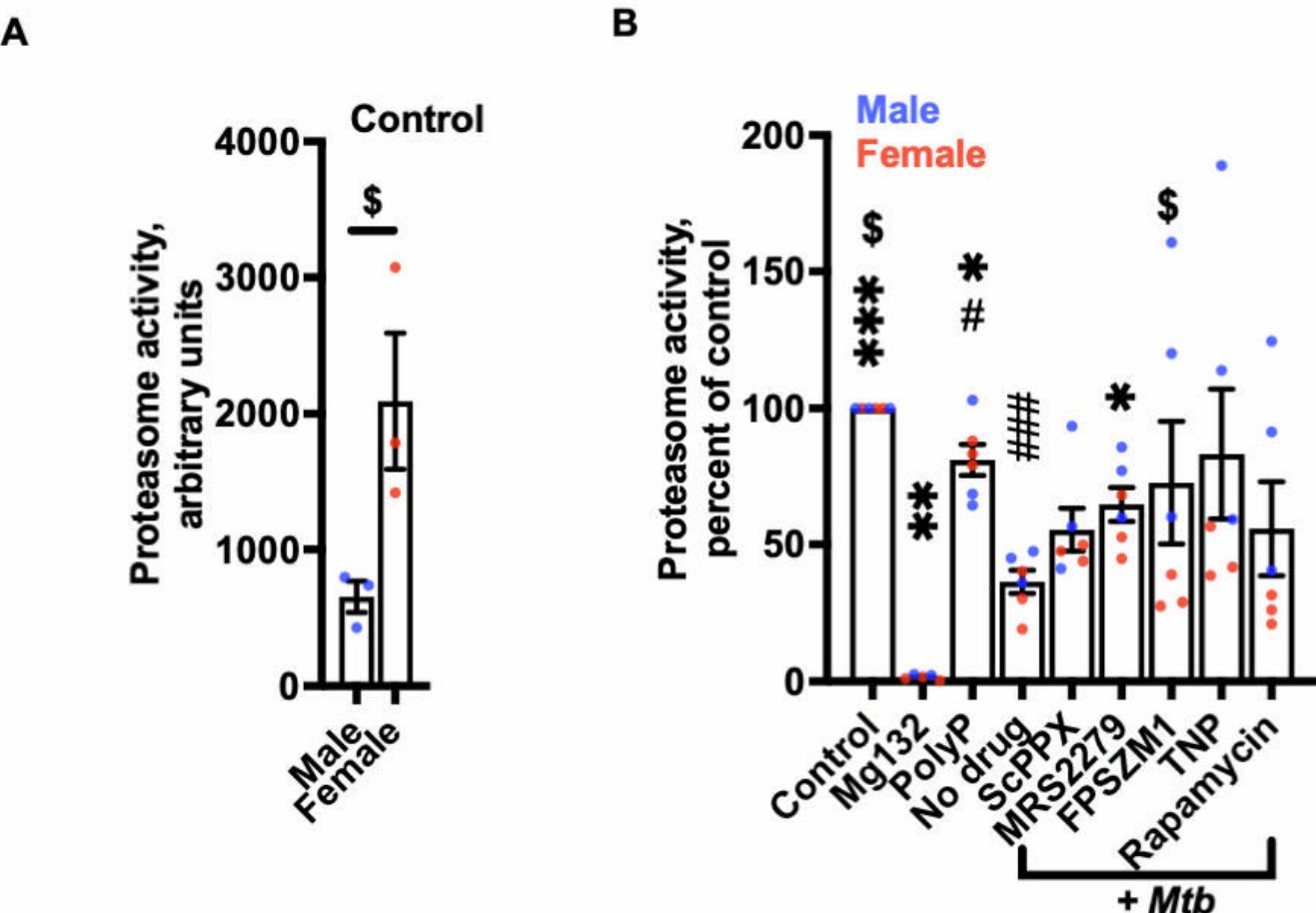

**FIG 4** Some inhibitors counteract *Mtb* inhibition of proteasome activity. (A and B) Proteasome activity of uninfected (control), 4 nM Mg132 or 15 µg/mL polyP treated, or *mtb* infected macrophages in the absence (No drug) or presence of 1,000 nM of the indicated inhibitor was assessed at 24 hours after phagocytosis. The proteasome inhibitor Mg132 was used as a control. The proteasome activity of the control was set to 100. All values are mean ± SEM of six (three females and three males) independent experiments. Male data points are shown in blue and female data points are shown in red. * or # $P < 0.05$; **$P < 0.01$; *** or ### $P < 0.001$ (one-way analysis of variance with Dunnett's test. *indicates compared to no drug; # indicates compared to control; $ indicates $P < 0.01$ female compared to male (unpaired *t*-test).

early events like phagosome maturation (60, 61). These findings suggest that IP6K may mediate other polyP-induced effects not directly related to phagosome acidification.

CD54 on macrophages enhances their phagocytic capabilities, particularly under inflammatory conditions induced by stimuli such as lipopolysaccharide. This enhancement is linked to the production of reactive oxygen species mediated by Toll-like receptor 4 signaling (53). The reduced expression of CD54 by *Mtb* and polyP suggests a mechanism by which these pathogens modulate macrophage polarization. However, the presence of ScPPX or inhibitors such as MRS2279, FPSZM1, TNP, and rapamycin restored CD54 expression, indicating that inhibiting macrophage components can counteract the immune evasion strategies of *Mtb* by promoting a proinflammatory macrophage phenotype.

CD206, also known as the macrophage mannose receptor, is involved in recognizing and binding to carbohydrate structures on the surface of *Mtb* (62–64). Once *Mtb* binds to CD206, this interaction limits phagosome-lysosome fusion, helping the bacteria to survive within macrophages (64). Both *Mtb* and polyP increased CD206 expression, but this effect was reduced by ScPPX, MRS2279, FPSZM1, TNP, and rapamycin. However, among these, only MRS2279 and FPSZM1 reversed the inhibitory effect of polyP on phagosome acidification, suggesting that polyP may suppress phagosome acidification

through a mechanism that does not require elevated CD206 levels or operates through partially distinct host responses.

The proteasome is crucial for the degradation of intracellular proteins into peptide fragments suitable for presentation by MHC Class I molecules, which are essential for the activation of cytotoxic T lymphocytes and the adaptive immune response (65, 66). Our findings suggest that *Mtb* may suppress proteasome activity in macrophages, potentially through mechanisms involving polyP. MRS2279 increased proteasome activity in *Mtb*-infected macrophages, whereas ScPPX, FPSZM1, TNP, and rapamycin did not. PolyP may trigger downstream pathways, such as purinergic signaling via P2Y1 receptors, that continue even after partial polyP degradation. Therefore, ScPPX's action may be too late to reverse signaling that has already been initiated by polyP. The lack of effect from TNP suggests that IP6K inhibition alone may be insufficient, possibly due to compensatory pathways such as those involving mTOR or Rho GTPases. Rapamycin may suppress proteasome function by inhibiting mTOR activity, which is known to regulate proteasome biogenesis (37). RAGE inhibition by FPSZM1 could also affect proteasome regulation indirectly, given its broader role in modulating inflammatory signaling (67, 68). It is also possible that the polyP-induced proteasome inhibition in macrophages proceeds via a distinct or noncanonical signaling mechanism not fully overlapping with the *Dictyostelium* pathway (31). The lack of strong inhibition in our system and the selective restoration by MRS2279 support this idea. These findings indicate that polyP may impair proteasome activity through multiple host pathways or *Mtb* may inhibit proteasome activity through polyP-independent mechanisms and that selective targeting of specific host factors may help restore proteasome function in *Mtb*-infected macrophages.

In conclusion, our findings suggest that targeting specific host proteins influenced by bacterial polyP can restore key macrophage functions, such as phagosome acidification and proteasome activity. This host-directed approach may enhance the clearance of intracellular pathogens including *Mtb*, *Legionella*, and *Listeria*. Future studies will be needed to determine how polyP levels and the effects of these inhibitors vary across different bacterial species and clinical isolates.

## MATERIALS AND METHODS

### Inhibitors

10 mM stocks of MRS2279 (Cat#2158, Tocris, Minneapolis, MN, USA) (34, 35), TNP (Cat#3946 Tocris) (38), FPS-ZM1 (Cat# 6237, Tocris) (36), Rapamycin (Cat# J62473, Alfa Aesar, Ward Hill, MA) (37), MBQ167 (Cat#HY-112842, MedChemExpress, Monmouth Junction, NJ) (39), Rhosin (Cat#5003, Tocris) (40), Y16 (Cat#SML0873, Sigma, St. Louis, MO) (41), and Exendin-4 (Cat#Enz-PRT111-0500, Enzo Life Sciences, Farmingdale, NY) (45, 46) were prepared in water or DMSO according to the manufacturer's instructions. Aliquots of 50 µL were stored at −20°C and were diluted in the macrophage culture media described below.

### Human cell culture

Human peripheral blood was collected from healthy volunteers who gave written consent, and with specific approval from the Texas A&M University human subjects institutional review board. Peripheral blood mononuclear cells (PBMCs) were purified as previously described (69). The PBMCs were cultured in RBCSG (Roswell Park Memorial Institute medium (12-167F, Lonza, Walkersville, MD, USA) containing 10% bovine calf serum (2,100–500, VWR Life Science Seradigm, Radnor, PA, USA) and 2 mM l-glutamine (Lonza), and where indicated containing 25 ng/mL human granulocyte-macrophage colony-stimulating factor (572903 GM-CSF) or 25 ng/mL human macrophage colony-stimulating factor (574804 M-CSF) (Biolegend, San Diego, CA, USA) at 37°C in a humidified chamber with 5% $CO_2$ in type 353219, 96-well, black/clear, tissue-culture-treated,

glass-bottom plates (Corning, Big Flats, NY, USA) with $10^5$ cells per well in 100 µL or type 353072, 96-well, tissue-culture-treated, polystyrene plates (Corning) with $10^5$ cells per well in 100 µL. At day 7, loosely adhered cells were removed by gentle pipetting, and fresh RBCSG containing GM-CSF or M-CSF was added to the cells to a final volume of 100 µL per well.

## Bacterial cell culture

The BSL-1 strain of *E. coli* K-12 (BW25113) (CGSC#7636) (70, 71), from the *E. coli* genetic stock center (70, 71), was grown at 37°C in Luria-Bertani (LB) broth (BD, Sparks, MD, USA) in a 50 mL conical tube (VWR) or on LB Agar on a type 25384-302 petri dish (VWR) at 37°C in a humidified incubator for 1 day. The attenuated (mc-ΔleuDΔpanCD) BSL-2 strain of *Mtb* (a derivative of the H37Rv strain) (32) (a gift from Dr. Jim Sacchettini, Department of Biochemistry and Biophysics, Texas A&M University, College Station, TX, USA) was grown as described (72) in Middlebrook 7H9 broth (BD, Sparks, MD, USA) in a type 89,039-656 50 mL conical tube (Falcon, VWR Life Science Seradigm) or on 7H10 agar (BD) at 37°C in a humidified incubator. Both *Mtb* media contained 0.5% glycerol (VWR), 0.05% Tween 80 (MP Biomedicals, Solon, OH), and the Middlebrook Oleic ADC Enrichment (BD). *Mtb* ΔleuDΔpanCD cultures (both liquid and agar plates) were additionally supplemented with 50 µg/mL leucine (VWR Life Science Seradigm) and 50 µg/mL pantothenate (Beantown Chemical, Hudson, NH). Liquid cultures were incubated in 50 mL conical tubes on a STR200-V variable angle tube rotator (Southwest Science, Roebling, NJ) for 1 to 2 weeks until the cell density reached log phase, and the agar plates were wrapped in plastic film to prevent desiccation and incubated for 3 to 4 weeks at 37°C in a humidified incubator. The BSL-2 strain of *Legionella* (*L. pneumophila* subsp. *pneumophila* Brenner et al.) (American Type Culture Collection (ATCC) 33153) was grown as described by ATCC (https://www.atcc.org/products/33153) in liquid 1099 CYE Buffered Medium or a solid 1099 CYE Buffered Medium at 37°C in a humidified incubator with 5% $CO_2$. for 3 days. The BSL-2 strain of *Listeria* (*L. monocytogenes* (et al.) Pirie (ATCC 19111) was grown as described by ATCC (https://www.atcc.org/products/19111) in a 44 Brain Heart Infusion Broth or on 44 Brain Heart Infusion Agar at 37°C in a humidified incubator for 2 days.

## Recombinant PPX and polyP

The plasmid for purifying *S. cerevisiae* exopolyphosphatase (ScPPX) was a kind gift from Michael Gray, University of Alabama at Birmingham, AL, USA (73). Recombinant ScPPX was purified as previously described in a protein purification protocol (74). 10 µg/mL of ScPPX was used to treat human macrophages in the assays. Long chain polyP (p700) (Cat#EUI002, Kerafast, Inc. Boston, MA, USA) was used in all assays. The polyP stock was prepared according to the manufacturer's instructions in sterile water to a concentration of 102 mg/mL polyP, and this stock was diluted in cultures to make 15 µg/ml polyP.

## Bacterial survival assay

To determine the effect of the inhibitors on the survival of *E. coli*, *Mtb*, *Legionella*, or *Listeria* in macrophages, human macrophages (from blood monocytes cultured with GM-CSF or M-CSF for 6 days) were infected with *E. coli*, *Mtb*, *Legionella*, or *Listeria*, in the absence or in the presence of 15 µg/mL polyP, 10 µg/mL ScPPX, or the indicated concentration of the inhibitor. At day 7, after removing loosely adhered cells as described above for PBMCs purification, 100 µL RBCSG (for *E. coli*, *Legionella*, and *Listeria* survival assay) or RBCSGLP (RBCSG containing 50 µg/mL leucine and 50 µg/mL pantothenate for *Mtb* survival assays) containing the indicated concentrations of the inhibitor without GM-CSF or M-CSF were added to macrophages in each well in type 353072, 96-well, tissue-culture-treated, polystyrene plates (Corning) and incubated for 30 minutes at 37°C. polyP, ScPPX, or the inhibitor when assayed, was present in all incubation steps up to the Triton lysis step. Meanwhile, 1 mL of *E. coli*, *Mtb*, *Legionella*, or *Listeria* from a log phase culture was washed twice with RBCSG (for *E. coli*, *Legionella,* and *Listeria*)

or RBCSGLP (for *Mtb*) without GM-CSF or M-CSF by centrifugation at 12,000 × *g* for 2 minutes in a microcentrifuge tube, resuspended in 1 mL of RBCSG (for *E. coli*, *Legionella*, and *Listeria*) or RBCSGLP (for *Mtb*), and the 600 nm optical density of 100 µL of the culture was measured with a Synergy Mx monochromator microplate reader (BioTek, Winooski, VT). One-hundred microliters of RBCSG (for *E. coli*, *Legionella*, and *Listeria*) or RBCSGLP (for *Mtb*) was used as a blank. The bacteria were diluted to an optical density of 0.5 (~0.33 × $10^7$ *Legionella*/mL; ~0.766  $10^7$ *Listeria*/mL; ~$10^7$*Mtb*/mL) in RBCSG (for *E. coli*, *Legionella,* and *Listeria*) or RBCSGLP (for *Mtb*). *Mtb* (~1 µL), *Legionella* (~3.3 µL), or *Listeria* (~1.3 µL) was added to macrophages in each well such that there were ~5 bacteria per macrophage considering ~20% of the blood monocytes converted to the macrophages in the presence of GM-CSF or M-CSF (75). The bacteria-macrophage co-culture plate was spun down at 500 × *g* for 3 minutes with a Multifuge X1R Refrigerated Centrifuge (Thermo Scientific, Waltham, MA, USA) to synchronize phagocytosis of the bacteria and incubated for 2 hours at 37°C. The supernatant medium was removed by gentle pipetting and was discarded. One-hundred microliters of phosphate buffered saline (PBS) warmed to 37°C was added to the co-culture in each well, cells were gently washed to remove un-ingested extracellular bacteria, the PBS was removed, and 100 µL of RBCSG (for *E. coli*, *Legionella*, and *Listeria*) or RBCSGLP (for *Mtb*) with M-CSF or GM-CSF containing 200 µg/mL gentamicin (Sigma, St. Louis, MO, USA) was added to the cells to kill the remaining un-ingested bacteria. After 2 hours, cells were washed twice with PBS as above to remove gentamicin. RBCSG (for *E. coli*, *Legionella*, and *Listeria*) or RBCSGLP (for *Mtb*) (100 µL) with M-CSF or GM-CSF was then added to the cells. After 4 and/or 48 hours of infection, macrophages were washed as above with PBS, the PBS was removed, and cells were lysed using 200 µL 0.1% Triton X-100 (Alfa Aesar) in PBS for 5 minutes at room temperature by gentle pipetting, and 20 and 100 µL of the lysates were plated onto agar plates (as described above for *Mtb* culture). The *Mtb*-containing agar plates were incubated for 3 to 4 weeks or until the *Mtb* colonies appeared, whereas *E. coli*-containing agar plates were incubated for 1 day (as described above for *E. coli* culture), *Legionella*-containing agar plates were incubated for 3 days (as described above for *Legionella* culture), and *Listeria*-containing agar plates were incubated for 2 days (as described above for *Listeria* culture). Bacterial colonies obtained from plating 20 and 100 µL lysates were manually counted, the number of viable ingested bacterial colonies per 20 and 100 µL lysates was calculated, and the number of viable ingested bacteria colony-forming units (CFUs) per mL of lysate was then calculated, which correspond to the number of viable ingested bacteria in ~2 × $10^5$ macrophages. To calculate the percent of control, CFU/mL $$$of the control of each independent experiment was considered 100%.

To determine the effect of the inhibitors on the survival of *Mtb* in human macrophages for more than 48 hours, macrophages were infected with *Mtb* as described above, in the absence or in the presence of the indicated concentrations of the inhibitor. The indicated concentrations of the inhibitor were then additionally added to the cells at 24 and 48 hours after *Mtb* infection. At 72 hours (3 days of infection), macrophages were lysed and plated onto agar (as described above for lysates from 4 and 48 hours). The agar plates were incubated for 3 to 4 weeks or until the *Mtb* colonies appeared. *Mtb* colonies obtained from plating 20 and 100 µL lysates were manually counted, the number of viable ingested *Mtb* colonies per 20 and 100 µL lysates was calculated, and the number of viable ingested *Mtb* CFU) per mL of lysate was then calculated, which correspond to the number of viable ingested *Mtb* in ~2 × $10^5$ macrophages. To calculate the percent of control, CFU/mL of the control was considered 100%.

## Bacterial growth assay

To investigate the effect of the inhibitors on bacterial growth, *Mtb*, *Legionella*, and *Listeria* bacteria were grown in a well of a 96-well, tissue culture-treated plate (# 353072, Corning) containing a final $OD_{600}$ of 0.1 in 200 µL of respective growth media as described above. OD600 was measured using a BioTek Synergy Mx monochromator

microplate reader. *Mtb*, *Legionella*, or *Listeria* bacteria were incubated in the absence or in the presence of 10 µg/mL ScPPX or the indicated inhibitor concentrations. Control wells contained medium with water or DMSO, which was similarly serially diluted in media. The plates were subsequently incubated in a container with humidity provided by wet paper towels at 37°C in a humidified incubator. The $OD_{600}$ of the cells was measured daily for 6 days (for *Mtb*) or at 12, 24, 48, 72, and 144 hours (for *Legionella*) or at 6, 12, 24, and 48 hours (for *Listeria*), and the bacterial growth curves were generated.

## Fluorescence microscopy

*Mtb* survival assays were performed in type 353219 96 well, black/clear, tissue culture-treated glass bottom plates (Corning) with $10^5$ cells/well in 100 µL as described above. In a control experiment, uninfected macrophages were incubated for 24 hours. At 24 hours, macrophages with ingested *Mtb* were fixed with 200 µL of 4% paraformaldehyde/PBS for 10 minutes. Cells were washed two times with 200 µL of PBS and permeabilized with 200 µL of 0.1% Triton X-100 (Alfa Aesar, Tewksbury, MA, USA) in PBS for 5 minutes. Macrophages were washed twice with 200 µL of PBS, blocked with 200 µL of 1 mg/mL type 0332 bovine serum albumin (VWR) in PBS for 1 hour, and washed once with 200 µL of PBS. Two-hundred microliters of 1:2,000 rabbit anti-*Mtb* antibody (Cat# OBT0947; Bio-Rad), 1:1,000 rabbit anti-CD54 (Cat#67836; Cell Signaling, Danvers, MA, USA), 1:1,000 rabbit anti-CD206 (Cat#91992; Cell Signaling) antibody in PBS/0.1% Tween 20 (Fisher Scientific, Pittsburgh, PA, USA) (PBST) was added to macrophages and incubated at 4°C overnight. Cells were washed three times with PBST and incubated with 200 µL of 1:500 Alexa 488 donkey anti-rabbit (711-546-152, Jackson Immunoresearch) or 1:500 Alexa 488 donkey anti-mouse (715-545-150, Jackson Immunoresearch) and 10 µg/mL of DAPI in PBST for 1 hour. Cells were washed three times with 200 µL of PBST, and 200 µL of PBS was then added to the well. Each washing step was done for 5 minutes, and all steps were performed at room temperature if not indicated otherwise. Images of macrophages were taken with a 100× oil-immersion objective or 40× objective on a Ti2 microscope (Nikon), and deconvolution of images was done using the Richardson-Lucy algorithm (76) in NIS-Elements AR software. Figures were prepared using Microsoft PowerPoint.

## Phagosome acidification assay

To determine the effect of inhibitors on polyP-mediated inhibition of phagosome acidification in human macrophages, macrophages derived from blood monocytes cultured with GM-CSF or M-CSF for 6 days in type 353219, 96-well, black/clear, tissue-culture-treated, glass-bottom plates (Corning) with $10^5$ cells per well in 100 µL (as described above) were incubated with a pH-sensitive pHrodo red labeled yeast (Cat# P35364, Thermo Fisher Scientific, Waltham, MA, USA) (yeast hereafter), and the acidification of the yeast containing phagosomes was monitored as previously described (25). pHrodo red labeled yeast is non-fluorescent outside the cell but fluoresces brightly red in acidic phagosomes (50). One-hundred nanometers of ConcanA (Cat#C9705-25UG, Sigma), a vacuolar-type H+-ATPase inhibitor that blocks phagosome acidification (51), was used as a positive control for inhibition of phagosome acidification.

## Proteasome activity assay

At 24 hours of phagocytosis, as described for the bacterial survival assay, cultures of macrophages in the absence or in the presence of 15 µg/mL polyP, 4 nM Mg132 (Cat#474787, Sigma), or *Mtb* without or with 1,000 nM inhibitor in type 353219, 96-well, black/clear, tissue-culture-treated, glass-bottom plates (Corning) were incubated with 100 µL/well of proteasome assay loading solution provided in a Proteasome Activity Kit (Cat#MAK172, Sigma, St Louis, MO). The plate was incubated in the dark for 1 hour, and proteasome activity was measured using a microplate reader following the manufacturer's instructions.

## Metabolic activity assay

Metabolically active cells transform the non-fluorescent blue dye resazurin into a fluorescent pink product resorufin (48). At 24 hours of phagocytosis, as described for the bacterial survival assay, macrophages in the absence or in the presence of *Mtb*, *Legionella*, *Listeria*, or inhibitor in 96-well, tissue culture-treated plates (# 353072, Corning) were incubated with prewarmed Deep Blue Cell Viability resazurin dye (Cat#424702, BioLegend, San Diego, CA, USA) to a final concentration of 10% in each well (77). Plates were incubated at 37°C for 12 hours. The fluorescence signal was then measured using a microplate reader following the manufacturer's protocol.

## Statistical analysis

Statistical analyses were performed using Prism 10 (GraphPad Software, Boston, MA, USA) or Microsoft Excel. $P < 0.05$ was considered significant.

## ACKNOWLEDGMENTS

The authors are thankful to the volunteers who donated blood and the phlebotomy staff at the Texas A&M Beutel Student Health Center.

This work was supported by the National Institutes of Health grant GM139486.

R.R. designed and performed experiments, analyzed data, and drafted the paper, and R.H.G. coordinated the study, revised the paper, and acquired funding.

## AUTHOR AFFILIATIONS

[1]School of Biological, Environmental, and Earth Sciences, The University of Southern Mississippi, Hattiesburg, Mississippi, USA
[2]Department of Biology, Texas A&M University, College Station, Texas, USA

## AUTHOR ORCIDs

Ramesh Rijal http://orcid.org/0000-0003-0498-1064
Richard H. Gomer http://orcid.org/0000-0003-2361-4307

## FUNDING

| Funder | Grant(s) | Author(s) |
| --- | --- | --- |
| National Institutes of Health | GM139486 | Richard H. Gomer |

## AUTHOR CONTRIBUTIONS

Ramesh Rijal, Conceptualization, Investigation, Methodology, Validation, Visualization, Writing – original draft, Writing – review and editing | Richard H. Gomer, Funding acquisition, Resources, Supervision, Validation, Writing – review and editing

## ADDITIONAL FILES

The following material is available online.

### Supplemental Material

**Supplemental figures (Spectrum02163-25-s0001.docx).** Figures S1 to S4.

### Open Peer Review

**PEER REVIEW HISTORY (review-history.pdf).** An accounting of the reviewer comments and feedback.

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
