## [Reviewer comments · Microbiology Spectrum]

Microbiology Spectrum

Pharmacological inhibition of host pathways enhances macrophage killing of intracellular bacterial pathogens

Ramesh Rijal and Richard Gomer

Corresponding Author(s): Ramesh Rijal, University of Southern Mississippi

Review Timeline:

Submission Date:	July 22, 2025
Editorial Decision:	September 30, 2025
Revision Received:	October 3, 2025
Accepted:	October 10, 2025

Editor: Travis Bourret

Reviewer(s): The reviewers have opted to remain anonymous.

Transaction Report:

DOI: <https://doi.org/10.1128/spectrum.02163-25>

Re: Spectrum02163-25 (**Pharmacological inhibition of host pathways enhances macrophage killing of intracellular bacterial pathogens**)

Dear Dr. Ramesh Rijal:

Thank you for the privilege of reviewing your work. Below you will find my comments, instructions from the Spectrum editorial office, and the reviewer comments.

I am pleased to inform you that your manuscript has been editorially accepted for publication. However, there are a few additional questions in the submission form that need to be answered before the final decision. Once these are completed, please return your submission so that I can move your paper forward to acceptance.

- Each figure must be uploaded as a separate, editable, high-resolution file (TIFF or EPS preferred), and any multipanel figures must be assembled into one file.

Sincerely,
Travis Bourret
Editor
Microbiology Spectrum

Reviewer #1 (Comments for the Author):

The work is well performed and has a broad impact. The authors have made a major effort to address all my concerns. The authors have removed few sections and discussed their results with better clarity in the discussion section of the revised manuscript.

Comments to Editor

It has been previously shown that pathogenic bacteria can block phagosome lysosome fusion to promote their intracellular survival. Previously, it has been shown that intracellular and extracellular polyP regulate the intracellular survival of *Mtb*. Previously, the authors have demonstrated that polyP requires mTOR complex protein Lst8, inositol hexakisphosphate kinase I6kA and Rho-GTPase RacE to facilitate the survival of ingested bacteria. In the present study, the authors show that pharmacological inhibition of these proteins potentiates the ability of macrophages to kill intracellular pathogens. The authors have used the following inhibitors: MRS2279 (inhibits P2Y1), FPSZM1 (inhibits RAGE), rapamycin (inhibits mTORC1), TNP (inhibits IP6K), MBQ167 (inhibits Rac/Cdc42), Rhosin and Y16 (inhibits RhoA) and Extensin-4 (inhibits human thioredoxin protein). The authors show that exposure of macrophages to MRS2279, TNP, FPSZM1 and rapamycin inhibited *Mtb* growth. In comparison, exposure to MBQ167, Rhosin, Y16 and Extensin-4 did not affect intracellular *Mtb* growth. MRS2279 and TNP also inhibited the growth of *Legionella* and *Listeria*. The authors also show that these compounds are unable to alter the metabolism of uninfected or infected macrophages or *Mtb* growth. As expected, the addition of exogenous polyP increased the number of ingested *E. coli* and exposure to inhibitors namely MRS2279, FPSZM1, TNP partially restored this defect. Also, it has been shown that polyP inhibits phagosome acidification and the authors show that this phenomenon is reversed in the presence of 1 μ M MRS2279 and FPSZMI. The authors also show that exposure to polyP decreased and increased CD54 and CD206 staining, respectively. Also, the authors show that this was reversed in the presence of inhibitors. Overall, the experiments have been well designed and performed but the defect/ restoration in the presence of inhibitors are minimal (mostly 2. 0-fold). The authors have not determined the activity of these small molecules in mice. The activity of these molecules should also be determined against drug-resistant bacteria in macrophages and mice. The combination experiments should also be performed in macrophages.

Comments to authors:

1. The figures have male and female data points in blue and red. However, these data points are not clear in the version I had.
2. Most of the intracellular killing assays are performed in GM-CSF or M-CSF stimulated BMDMs. The authors should repeat these assays in non-stimulated cells i.e. BMDMs or a macrophage cell line.
3. In Fig. 2B, the restoration seen in the presence of inhibitors is minimal.
4. The statistical significance is missing in most figures.
5. The authors show that exposure to MRS2279 and FPSZMI can restore the effect of polyP on phagosome acidification. However, this is a very marginal restoration. The authors also did not see any effect in the presence of TNP? How do the authors explain this?
6. Previously, it has been shown that polyP inhibits proteasome activity in *Dictyostedium* and this inhibition requires mTOR protein Lst8, RacE, I6kA, AdC and Gr1D. In the presence study, inhibition of proteasome activity in the presence of polyP is very minimal. Also, the reversal of proteasome inhibition by polyP was only observed in the presence of MRS2279. Further, no restoration was seen in the presence of other other inhibitors? How do the authors explain these observations?

7. Fig. 4C and 4D: The author shows that incubation with polyP did not significantly affect LC3-II levels. However, MRS2279, TNP or rapamycin increased LC3-II levels. This looks to be a polyP independent effect. The data should be removed.
8. The major challenge in the field is the emergence of drug-resistant strains and identifying small molecules that can shorten the treatment duration. The authors have not tested of the small molecules in mice model.
9. Have the authors tested these molecules in combination with known TB drugs such as INH, Rif etc.

Comments to authors

The work is well performed and has a broad impact. The authors have made a major effort to address all my concerns. The authors have removed few sections and discussed their results with better clarity in the discussion section of the revised manuscript.

I believe that the work is now acceptable for publication.

1 Re: mBio01070-25 (**Pharmacological inhibition of host pathways enhances**
2 **macrophage killing of intracellular bacterial pathogens**)

3
4 Dear Dr. Rijal:

5
6 We have completed our review of your manuscript, and I regret to inform you that we will
7 not be able to accept it for publication in mBio. This decision reflects the priorities of mBio
8 and the disposition of this particular review; it is not meant to imply that the manuscript is
9 unsuitable for publication elsewhere.

10
11 While this manuscript may not be a good fit for mBio, please note that you do have the
12 option to transfer this manuscript to Microbiology Spectrum. Spectrum is an ASM open-
13 access journal that publishes technically sound, primary research across the entire range
14 of microbial sciences and allied fields. As your article was reviewed at mBio, you can
15 transfer the paper along with the reviews and reviewer identities to Spectrum. The
16 Spectrum Editors will carefully assess your response to technical concerns raised in peer
17 review in their decision-making process. **Please note that Spectrum only publishes**
18 **primary research and does NOT currently publish or consider case studies,**
19 **reviews, meta-reviews, commentaries, opinions/hypotheses, perspectives, or**
20 **minireviews.**

21
22 **If you would like to transfer the manuscript and the mBio reviews to Spectrum,**
23 **please use the link below.** Note that the transfer link will be visible only in the decision
24 letter sent to the corresponding author. [https://mbio.msubmit.net/cgi-](https://mbio.msubmit.net/cgi-bin/main.plex?el=A5FE7CdOd7A6GhLI4X2C9ftd08GcNAvIQIae55CREyjUvQZ)
25 [bin/main.plex?el=A5FE7CdOd7A6GhLI4X2C9ftd08GcNAvIQIae55CREyjUvQZ](https://mbio.msubmit.net/cgi-bin/main.plex?el=A5FE7CdOd7A6GhLI4X2C9ftd08GcNAvIQIae55CREyjUvQZ). Please
26 note that each ASM journal is editorially independent; transferring your manuscript
27 constitutes a new submission. Your manuscript will be subject to the editorial decisions
28 of the receiving journal.

29
30 Enclosed are the comments generated during the review. I hope they will be useful to
31 you.

32
33 Thank you for submitting to mBio. We hope that you will consider us in the future.

34
35 Sincerely,
36 Arturo Casadevall
37 Editor in Chief, mBio
38 (Signing for the editors)

39

40 Editor comments (if any):

41 Two expert reviewers noted the importance of the topic of the manuscript and made
42 substantive and detailed comments regarding the described studies. Each noted several
43 new experiments that would need to be performed to validate the current studies, as well
44 as some presented data regarding inhibitors without function that did not validate the
45 authors' conclusions. Also, the effects of the inhibitors that did have some activity was
46 noted to be "minimal", and the actual mechanistic effects of some of the most important
47 small molecules in mice not yet been demonstrated, making many of the inferences
48 indirect.

49

50 **Reviewer #1:**

51 Bacterial Polyphosphate (PolyP) promotes intracellular survival of bacteria by, among
52 other things, inhibiting acidification of phagosomes and lysosomal activity in
53 macrophages. The authors of this manuscript previously found that that bacterial PolyP
54 hinders the ability of *Dictyostelium discoideum* to kill intracellular bacteria via the RAGE,
55 P2Y1, mTOR and I6kA pathways. In this interesting and comprehensive study they now
56 extend this line of investigation to human macrophages (MDM). They find that treatment
57 with a recombinant polyphosphatase that degrades polyP, or with inhibitors of these host
58 signalling pathways, improves the ability of MDM to kill *M. tuberculosis* (*Mtb*), *Legionella*
59 *pneumophila* and, to a lesser extent, *Listeria monocytogenes*. The addition of exogenous
60 polyP to macrophages inhibited macrophage anti-microbicidal activities such as
61 acidification of phagosomes and killing of intracellular bacteria. Some of the inhibitors
62 reversed this.

63

64 Comments:

65 1. All of the work with *Mtb* was carried out with the attenuated $\Delta\text{leuD}\Delta\text{panCD}$ BSL2
66 auxotroph of H37Rv. While the auxotroph recapitulates many of the same effects on
67 host innate immune cells as its virulent BSL3 counterpart, it is more susceptible to the
68 effects of acid stress which results in reduced replication (Mouton et al, PMID:
69 31481950). Consequently, this strain may also be more susceptible to the effects of
70 inhibitors that improve phagosome acidification and it would be important to confirm
71 at least some of the CFU results with the inhibitors in macrophages infected with
72 virulent H37Rv.

73 We started from our work identifying *Dictyostelium discoideum* mutants that potentiate
74 the killing of ingested *E. coli* in the presence of exogenous polyP. These mutants lack
75 a variety of signal transduction pathway components. In this report, we show that
76 inhibitors of orthologues of some of the identified pathway components potentiate the
77 ability of human macrophages to kill 3 different species of bacteria. One of these 3
78 bacterial species was the 'BSL-2 plus' attenuated *Mtb* strain. We now mention this
79 clearly in the last paragraph of the Introduction. Although we agree that it would be
80 nice to study the BSL-3 strain of *Mtb*, we think that this is well beyond the scope of
81 this report.

82

83 2. Pharmacological inhibitors of various pathways were used in the study, they may have
84 off target effects on different pathways and processes in macrophages so at least

85 some of the results should be confirmed by other means, for example using
86 neutralising antibodies to the cell surface receptors RAGE and P2Y1.

87 We agree. The inhibitors may affect other pathways beyond P2Y1 or RAGE, and the
88 exact mechanisms remain unclear. We now mention this in the Discussion. We plan
89 to include neutralizing antibodies and CRISPR knockdown of specific targets in follow-
90 up studies.

- 91
- 92 3. The evidence for a role for autophagy in PolyP-related pathways (Figure 4) is weak.
93 The changes in LC3-II levels are minimal even with the inhibitors. An increase in LC3-
94 II levels on a western blot is not necessarily indicative of increased autophagic flux,
95 the opposite is often the case. In addition, there was no significant increase in LC3-II
96 levels with Mtb infection - others have shown increased LC3-II in virulent Mtb-infected
97 macrophages that is associated with a block in autophagy (e.g. Ge et al (2021) PMID:
98 34092182; Petruccioli et al (2012) PMID: 22457295).

99 We agree that LC3-II alone does not indicate autophagic flux. We removed these data
100 from the manuscript. In future work, we plan to include additional markers such as p62
101 and flux assays to better assess autophagy.

- 102
- 103 4. It would be necessary to look concurrently at levels of LC3-II and p62 to be able to
104 determine whether autophagic flux is being induced or prevented by the inhibitors.
105 Controls involving treatment with bafilomycin for the last few hours of the incubation
106 compared to no treatment would also be helpful in determining whether the inhibitors
107 are truly increasing autophagic flux.

108 As above, these experiments will be part of planned follow-up work. For this study, the
109 LC3-II data and discussion have been removed.

- 110
- 111 5. Figures: Individual data points showing male and female subjects are not visible on
112 any of the graphs - is this a software problem? Also check the labels on X and Y axis
113 - in many cases they make no sense: words are misspelt and/or superimposed so as
114 to render them uninterpretable. Symbols indicating statistical significance are missing
115 from many of the graphs.

116 This was due to PDF conversion problems. All figures are now corrected and clearly
117 show male/female data points, readable labels, and statistical markers.

- 118
- 119 6. It would be helpful to readers to include the target of the inhibitors in the figure titles
120 or legend.

121 We added inhibitor targets to figure legends where appropriate.

- 122
- 123 7. Figure 3 - were the fluorescence intensity values corrected for cell numbers?

124 Yes. The figure legend now states: "mean fluorescence intensity per cell was
125 measured."

126
127

128 **Reviewer #2:**

129 Comments to authors:

- 130 1. The figures have male and female data points in blue and red. However, these data
131 points are not clear in the version I had.
132 This was a PDF conversion issue at mBio, and we will carefully check the uploaded
133 figures.
134
- 135 2. Most of the intracellular killing assays are performed in GM-CSF or M-CSF stimulated
136 BMDMs. The authors should repeat these assays in non-stimulated cells i.e. BMDMs
137 or a macrophage cell line.
138 We used GM-CSF and M-CSF differentiated macrophages to mimic inflammatory and
139 tissue-resident phenotypes. We think that testing in two physiologically relevant types
140 of human macrophages is sufficient for this report, although we do plan to test cell
141 lines and non-stimulated macrophages in future studies.
142
- 143 3. In Fig. 2B, the restoration seen in the presence of inhibitors is minimal.
144 This is however statistically significant. We added text in the Discussion
145 acknowledging the modest effect and discussing potential explanations.
146
- 147 4. The statistical significance is missing in most figures.
148 This was a PDF conversion issue at mBio, and we will carefully check the uploaded
149 figures.
150
- 151 5. The authors show that exposure to MRS2279 and FPSZMI can restore the effect of
152 polyP on phagosome acidification. However, this is a very marginal restoration. The
153 authors also did not see any effect in the presence of TNP? How do the authors
154 explain this?
155 We added a new paragraph in the Discussion explaining that TNP likely affects
156 downstream signaling, and may not influence phagosome acidification directly.
157
- 158 6. Previously, it has been shown that polyP inhibits proteasome activity in Dictyostedium
159 and this inhibition requires mTOR protein Lst8, RacE, I6kA, AdC and GrID. In the
160 presence study, inhibition of proteosome activity in the presence of polyP is very
161 minimal. Also, the reversal of proteosome inhibition by polyP was only observed in the
162 presence of MRS2279. Further, no restoration was seen in the presence of other other
163 inhibitors? How do the authors explain these observations?
164 We now address this in the Discussion. The selective effect of MRS2279 suggests a
165 unique pathway through P2Y1. Other inhibitors may act through different axes or be
166 insufficient on their own. We will test combinations and mechanistic interactions in
167 follow-up work.
168
- 169 7. Fig. 4C and 4D: The author shows that incubation with polyP did not significantly affect
170 LC3-II levels. However, MRS2279, TNP or rapamycin increased LC3-II levels. This
171 looks to be a polyP independent effect.
172 These data and associated text have been removed for clarity. We plan to examine
173 autophagy in future experiments.

174

175 8. The major challenge in the field is the emergence of drug-resistant strains and
176 identifying small molecules that can shorten the treatment duration. The authors have
177 not tested of the small molecules in mice model.

178 Correct. In vivo work is planned for future studies and is outside the scope of this in
179 vitro-focused manuscript.

180

181 9. Have the authors tested these molecules in combination with known TB drugs such
182 as INH, Rif etc.

183 We agree this is important, but we think that this is beyond the scope of this report,
184 and would best be done in the BSL-3 strain of *Mtb*.

Re: Spectrum02163-25R1 (**Pharmacological inhibition of host pathways enhances macrophage killing of intracellular bacterial pathogens**)

Dear Dr. Ramesh Rijal:

Your manuscript has been accepted, and I am forwarding it to the ASM production staff for publication. Your paper will first be checked to make sure all elements meet the technical requirements. ASM staff will contact you if anything needs to be revised before copyediting and production can begin. Otherwise, you will be notified when your proofs are ready to be viewed.

Sincerely,
Travis Bourret
Editor
Microbiology Spectrum